

# 120 years of sea-ice cover on the Northeast Greenland continental shelf: a biomarker and observational record comparison

Joanna Davies[1], Kirsten Fahl[2], Matthias Moros[3], Alice Carter-Champion[1,4], Henrieka Detlef[1], Ruediger Stein[2,5,6] Christof Pearce[1], Marit-Solveig Seidenkrantz[1]

[1]Department of Geoscience, Arctic Research Centre, and iClimate, Aarhus University, Aarhus, Denmark
[2]Alfred Wegener Institute (AWI) Helmholtz Centre for Polar and Marine Research, Bremerhaven, Germany
[3]Leibniz Institute for Baltic Sea Research Warnemünde, Seestraße 15, DE-18119 Rostock, Germany
[4]Centre for Quaternary Research, Department of Geography, Royal Holloway, University of London
[5]Faculty of Geosciences and Center for Marine Environmental Sciences (MARUM), University of Bremen, Bremen, Germany
[6]Frontiers Science Center for Deep Ocean Multispheres and Earth System & Key Laboratory of Marine Chemistry Theory and Technology, Ocean University of China (OUC), Qingdao, China

*Correspondence to:* Joanna Davies (joannadavies@geo.au.dk)

**Abstract.** This study reconstructs recent changes (ca. 120 years) in sea-ice cover, using biomarkers ($IP_{25}$ and phytoplankton sterols) from three sediment cores located in a transect across Belgica Trough, on the Northeast Greenland continental shelf. These results are evaluated using instrumental and historical data from the same region and time period. Over the entire 120-year study period, $IP_{25}$ concentrations are highest at the inner shelf (site 90R) and decrease towards the mid-shelf (site 109R), with lowest values found at the outer shelf (site 134R). The $PIP_{25}$ index yields the highest sea-ice cover at sites 109R and 90R and lowest at 134R, in agreement with observational records. A decline in sea-ice cover, identified visually and using change point analysis, occurs from 1971 in the observational sea-ice data at sites 90R and 109R. A change in sea-ice cover occurs in 1984 at site 134R. Sea-ice cover in these years aligns with an increase in sterol biomarkers and $IP_{25}$ at all three sites and decline in the $PIP_{25}$ index at sites 90R and 134R. The outcomes of this study support the reliability of biomarkers for sea-ice reconstructions in this region.

## 1 Introduction

The decline in Arctic sea ice provides stark evidence that our climate is changing. In the last four decades, the extent of summer sea ice in the Arctic has decreased by around 44% (Perovich et al., 2020). This is affecting global climate, through changes in albedo, freshwater flux, and surface heat exchange (Sévellec et al., 2017; Thomas, 2017). The decline in sea-ice cover is attributed to warming air (Ananicheva et al., 2011) and ocean temperatures (Polyakov et al., 2011), and to changes in radiative (Kapsch et al., 2016) and wind forcings (Ogi et al., 2010). Yet, there is still uncertainty in the depiction of sea-ice cover and seasonality in climate models, due, in part, to the relatively short length of instrumental records; specifically, continuous satellite data of sea ice that extends back only to 1979, and some historical records, which extend back to 1850 (Walsh et al., 2019). In the absence of such records, we rely on proxies to reconstruct changes on longer timescales.





Biogenic and geochemical proxies have been developed to reconstruct sea-ice cover (Stein et al., 2012; Belt and Müller, 2013;
de Vernal et al., 2013). The mono-unsaturated highly branched isoprenoid (HBI) $IP_{25}$ is produced solely by Arctic sea ice dwelling
diatoms during the spring bloom and deposited in sediments below when the sea ice melts (Belt et al., 2007; Belt, 2018, 2019).
Together with biomarkers of open water dwelling phytoplankton (e.g., brassicasterol and dinosterol), these lipids can be used to
reconstruct semi-quantitative estimates of sea-ice cover by calculating the $PIP_{25}$ index; this differentiates perennial sea-ice cover
and open water as both can be characterised by an absence of $IP_{25}$ (Müller et al., 2011).

Sea-ice biomarkers (i.e., $IP_{25}$ and $PIP_{25}$) have been tested on surface samples by comparing reconstructions to instrumental
data (Müller et al., 2011; Navarro-Rodriguez et al., 2013; Xiao et al., 2015b; Kolling et al., 2020), and subsequently used
extensively for reconstructions of Holocene sea-ice conditions throughout the Arctic (Müller et al., 2012; Stein et al., 2017; Syring
et al., 2020b, a; Georgiadis et al., 2020; Detlef et al., 2021; Jackson et al., 2021; Detlef et al., 2023). Furthermore, some high-
resolution sea-ice biomarker studies have been carried out for the last millennium, characterised by centennial cold and warm
phases such as the Medieval Climate Anomaly and the Little Ice Age (Massé et al., 2008; Kolling et al., 2017). However, far fewer
studies have undertaken high-resolution reconstructions that encapsulate changes over the last century and compared these results
with instrumental datasets (Vare et al., 2010; Alonso-García et al., 2013; Weckström et al., 2013; Cormier et al., 2016; Pieńkowski
et al., 2017; Kim et al., 2019). Currently, there are no such high-resolution records of sea-ice cover from Northeast Greenland.

The upper parts of sediment cores often span the redox sedimentary boundary; this makes the degradation of sterols and HBIs
more likely (Rontani et al., 2018). Unfortunately, this part of the record frequently coincides with anthropogenic climate change,
where dramatic changes in sea-ice cover have occurred. Thus, disentangling degradation signals from recorded environmental
changes is important to improve the reliability of reconstructions of past sea-ice cover. Such high-resolution studies can be used
to address this issue, together with data from the ratio of two common phytoplankton sterols: epi-brassicasterol and 24-
methylenecholesterol. This has previously been used to investigate autoxidation processes, which can affect the preservation of
$IP_{25}$ (Rontani et al., 2018).

Historical data from the Nordic Seas indicate changes in the position of the sea-ice margin over recent centuries (Divine and
Dick, 2006), with an even stronger shift in perennial (multi-year) to seasonal sea-ice cover across the Arctic marginal seas in recent
decades (Nghiem et al., 2007; Onarheim et al., 2018; Wang et al., 2022); yet these latter changes are neither seasonally nor
regionally uniform (Årthun et al., 2021). Northeast Greenland is an area characterised by several sea ice types and features; it is
thus a region of interest to understand the impact of climate changes on sea-ice extent. These features include land-fast sea ice
(hereafter 'fast ice'), seasonal sea ice and the Northeast Water (NEW) polynya. Its location also means that it is influenced by sea
ice exported from the Arctic Ocean through the Fram Strait. This results in the presence of multi-year sea ice in the region, on the
outer shelf. Due to its extensive sea-ice cover, this area remained relatively under-sampled until recently. It is also an area that has
been subject to retreat of sea ice in recent decades. Consequently, Northeast Greenland is an ideal location to test time series
comparisons of sea-ice proxies versus observational records, both to further develop the proxies and to investigate whether there
have been previous periods of sea ice loss during the last century.

In this study, we present the first high-resolution biomarker reconstructions of sea-ice conditions over the last ca. 120 years
from the Northeast Greenland shelf based on analyses of three marine sediment cores that span a transect from the inner to outer



Belgica Trough (Fig. 1). This data is compared with satellite and historical data to evaluate biomarkers for reconstructing sea-ice

cover in this region.

## 2 Regional Setting

Sea ice along the Northeast coast of Greenland can broadly be divided into four zones, seen from the shoreline towards the open sea: 1) fast sea, which forms along the coast and remains stationary throughout the winter months, 2) a transitional zone, forming the boundary between the fast ice and the pack ice zone, the 3) the pack ice zone, which is composed of ice originating in the

Arctic Ocean and subject to movement via both wind and ocean currents, and finally 4) the marginal ice zone which defines the area between the pack ice and open water (Wadhams, 1981; Pedersen et al., 2010). Areas of open water are common within the transitional zone, where the mobile pack ice moves along the stationary fast ice; this includes well-defined polynyas (Pedersen et al., 2010).

The Norske Øer Ice Barrier (NØIB) is an area of perennial, fast ice that acts as a buttress to marine terminating glaciers in

Northeast Greenland, reducing calving at the glacier front (Fig. 1C; Reeh et al., 2001). The formation of fast ice begins in the shallow areas of the Belgica Bank, with grounded ridge keels and icebergs keeping the drift ice in place (Hughes et al., 2011). The fast ice is made up of old drift ice carried southwards from the Arctic Ocean by the East Greenland current and held together with newly formed ice (Hughes et al., 2011). The NØIB was first observed in 1938 (Koch, 1945). Whilst there has been considerable variation in the extent of the NØIB, in the past its breakup was a rare event, estimated to occur once every about 50 years (Higgins,

1989). However, the NØIB has been disintegrating almost every summer since 2000, likely driven by warming oceans and atmosphere (Sneed and Hamilton, 2016); this has resulted in increased calving of nearby marine terminating glaciers (Reeh et al., 2001). Furthermore, direct observations of the NØIB made between 2012 and 2013 indicate that high ocean heat fluxes, combined with thick snow cover resulted in net loss of ice at the ice bottom (Wang et al., 2020).

The Northeast Greenland Ice Stream (NEGIS) drains approximately 12% of the ice sheet interior, via two marine terminating

glaciers: Nioghalvfjerdsfjorden Glacier (NG) and Zachariae Isstrøm (ZI) (Fahnestock et al., 2001; Hvidberg et al., 2020; Khan et al., 2014; Mouginot et al., 2015). Due to its topographic setting, the NEGIS is particularly vulnerable to climate change and retreated and thinned rapidly in recent decades (Khan et al., 2014; Mouginot et al., 2015). Modelling estimates suggest that the marine based sectors of the NEGIS will contribute between 13.5 and 15 mm to sea-level rise by 2100 (Khan et al., 2022). Whilst 79NG has a more stable grounding line than ZI, modelling studies indicate that loss of the 79NG ice shelf would cause the

grounding line to retreat by around 10 km (Mayer et al., 2018). Direct observations have also linked reduced sea-ice concentrations to glacier front calving between 2002 and 2004 (Khan et al., 2014). Thus, accurate depictions of past sea-ice cover in Northeast Greenland are important to understand the processes driving the stability of these glaciers in the future.



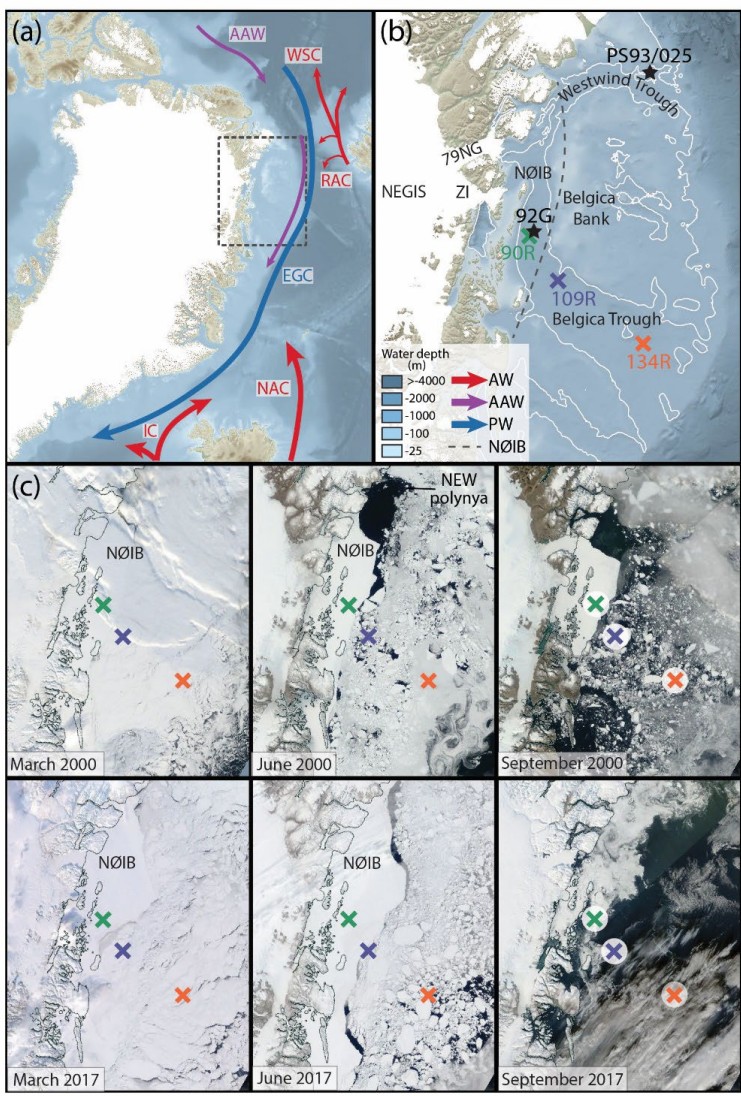

**Figure 1:** (a) an overview of the study site showing key ocean circulation patterns, namely warm currents (red; North Atlantic Current (NAC),
Irminger Current (IC), Return Atlantic Current (RAC), West Spitsbergen Current (WSC)), Arctic Atlantic Water (AAW; purple) and East
Greenland Current (blue). (b) A close up of the Northeast Greenland continental shelf, showing the location of the three sediment cores used in
this study: 90R (green), 109R (dark purple) and 134R (orange). Gravity core 92G is shown (black star) and sediment core PS93/025 (black star;
Syring et al., 2020a). The ocean bathymetry comes from GEBCO and ice velocity data from Sentinel- SAR data from 2019-2020 (Nagler et al.,
2015). The glaciers that make up the Northeast Greenland Ice Stream (NEGIS) are shown: Zachariæ Isstrøm (ZI) and Nioghalvfjerdsfjorden
Glacier (79NG). The approximate location and extent of the Norske Øer Ice Barrer (NØIB) is marked (black dotted line). (c) satellite data showing
sea-ice conditions from (top) 16 March (left), 6 June (middle) and 4 September (right) from 2000 and from (bottom) 15 March (left), 3 June
(middle), and 19 September 2017 (right), respectively. The location of the Northeast Water (NEW) polynya and NØIB is marked. The core
locations are shown as coloured crosses. Images courtesy of the NASA Worldview (https://worldview.earthdata.nasa.gov/).



## 3    Materials and Methods

### 3.1  Sampling

Three Rumohr cores, DA17-NG-ST08-090R, DA17-NG-ST10-109R and DA17-NG-ST12-134R, henceforth 90R, 109R and 134R, respectively, are used in this study (Fig. 1b; Table 1). All cores were collected from the Belgica Trough, which crosses the Northeast Greenland shelf from NW-SE. For all cores, the sediment-water interface was undisturbed at the time of sampling (clear sea water was preserved in the tube above the sediment), showing that the sediment surface was most likely intact. Gravity core
DA17-NG-ST08-092G, henceforth 92G, (Fig. 1b) located close to 90R is also used to assist in the chronology. All cores were collected on board the Danish research vessel *DANA* in 2017 as part of the NorthGreen2017 Expedition (Seidenkrantz et al., 2018). After collection, the cores were stored at 3 °C, at Aarhus University, where they were split and sampled for analysis.

**Table 1:** Location and water depth of the three Rumohr cores analysed, and the gravity core. The coordinates for the corresponding historical
sea-ice data (obtained from Walsh et al., 2019) are also shown.

| Core | Short name | Lat. (core) | Long. (core) | Lat. (sea-ice data) | Long. (sea ice-data) | Water depth (m) |
|---|---|---|---|---|---|---|
| DA17-NG-ST08-090R | 90R | 78.500 | -17.307 | 78.625 | -17.375 | 595 |
| DA17-NG-ST10-109R | 109R | 77.95 | -15.493 | 77.875 | -15.375 | 503 |
| DA17-NG-ST12-134R | 134R | 77.125 | -10.663 | 77.125 | -10.625 | 501 |
| DA17-NG-ST08-092G | 92G | 78.501 | 17.279 | NA | NA | 583 |

### 3.2  Sampling

Sediment samples were taken at 0.5 cm resolution in the upper parts of the three cores (10 cm for 109R, 7.5 cm for 134R, and 8 cm for 90R); one further sample was taken at 14 cm in all three sediment cores. These samples were weighed, freeze-dried, and homogenised. Between 2.5 and 5 g of freeze-dried sediment was taken for biomarker analysis, for chronological constraint (100
mg) and between 15 and 20 mg was taken for total organic carbon (TOC), total nitrogen (TN) and $\delta^{13}C_{org}$ analyses.

### 3.3  X-ray fluorescence

Sediment cores 92G and 90R were scanned by X-ray fluorescence (XRF) with and with an ITRAX scanner equipped with a Molybdenum X-ray tube at 30 kV and 30 mA to record the elemental spectra at 0.2 mm resolution, at the Department of Geoscience, Aarhus University.

### 3.4  Grain size analysis

Grain size analysis was undertaken on sediment cores 92G and 90R using a Sympatic HELOS laser diffractometer, equipped with a Quixel wet disperser, 6 mm cuvette and R4 lens, at 1 cm at the Department of Geoscience, Aarhus University. Peptizer solution



(10 ml (NaPO3)6 15%) was first added to the laser diffractometer and a reference measurement undertaken, before the sample was measured. Grain sizes were grouped into three fractions: sand (>63 mm), medium to coarse silt (15.6-63 mm), very fine to fine silt

(3.9-15.6 mm) and clay (0.6-3.9 mm).

### 3.5 Organic bulk sediment parameters

The freeze-dried and homogenized sediment (15-20 mg) was added into silver capsules with 30 µL Silex water. The samples were put in a microtiter plate, together with a beaker with concentrated HCl, and placed in desiccator in a vacuum until the acid boiled (6-24 h). This removes any carbonates from the sediment sample. Samples were then dried gently in an oven at 60 °C and the

capsules were then closed and analysed for TOC on an elemental analyser connected in continuous flow mode to an isotope ratio mass spectrometer (Delta V, Thermo Scientific). This analysis was undertaken at the University of Copenhagen, Denmark.

### 3.6 Biomarker analysis

Analysis was undertaken at the Alfred Wegener Institute for Polar and Marine Research, Bremerhaven. Between 2.5 and 5 g of the freeze-dried and homogenized sediment was used for each sample. Internal standards were added to each sample prior to extraction,

7-hexylnonadecane (7-HND, 0.076 µg/sample), 9-octylheptadec-8-ene (9-OHD, 0.01µg/sample), 5α-androstan-3β-ol (Androstanol, 1.07 µg/sample) and Squalane (0.32 µg/sample), for quantification. Extraction was undertaken using dichlormethane/methanol (DCM/MeOH, 2:1), followed by ultrasonication (15 min.) and centrifugation (3 min); this was undertaken three times. The solvent was then removed under a slow stream of nitrogen. Hexane was then added to the extract and column chromatography undertaken.

Column chromatography was undertaken using $SiO_2$ gel, with 5 ml of n-hexane and 9 ml of ethylacetate/n-hexane. Before measuring, the polar fraction, containing sterols, was silylated using 200 µl of bis-trimethylsilyl-trifluoroacetamide (BSTFA) at 60 °C for 2 hours. The HBIs were measured using a gas chromatograph Agilent Technologies 7890B GC system (30m DB-1MS column, 0.25 mm i.d., 0.25 µm film thickness) coupled to a mass spectrometer Agilent 5977A MSD (70 eV constant ionization potential, Scan 50–550 m/z, 1 scan/s, ion source temperature 230 °C, Performance Turbo Pump) with the temperature program:

60 °C (3 min), 150 °C (heating rate: 15 °C/min), 320 °C (heating rate: 10 °C/min) and 320 °C (15 min, isothermal). Sterols were measured with a GC Agilent 6850 GC (30m DB-1MS column, 0.25 mm i.d., 0.25 µm film thickness) coupled to an Agilent 5975C VL MSD with the temperature sequence: 60 °C (2 min), then 150 °C (heating rate: 15 °C/min), 320 °C (heating rate: 3 °C/min) and 320 °C (20 min isothermal).

    The identification and quantification of biomarkers was done in ChemStation and is based on individual retention times and

mass spectra: The highly branched isoprenoid (HBI) and sterols were identified: $IP_{25}$ at m/z 350, HBI II at m/z 348, HBI III (both isomers) at m/z 346, epi-brassicasterol (24-methylcholesta-5,22-dien-3β-ol) at m/z 470, dinosterol (4α,23,24-Trimethyl-5α-cholest-22E-en-3β-ol) at m/z 500, sitosterol (24-ethylcholest-5-en-3β-ol) m/z 486, and campesterol (24-methylcholest-5en-3β-ol) at m/z 472. 24-Methylenecholesterol (24-methylcholesta-5,24(28)-dien-3 β-ol) at m/z 470 was also identified for biodegradation rate calculations, see below (Rontani et al., 2018). All sterols were quantified as trimethylsilyl ethers. External calibration curves

were applied, and specific response factors were applied to balance the different responses of molecular ions of the analytes and



the molecular/fragment ions of the internal standards. For more details, please refer to (Fahl and Stein, 2012). The identified peaks were integrated and compared to the integrated peak area of the added internal standards. The data was normalised using the TOC of samples and freeze-dried sediment weights.

The $PIP_{25}$ index was calculated using the following Eq. (1) (Müller et al., 2011):

$$PIP_{25} = \frac{IP_{25}}{(IP_{25} + (phytoplankton\ marker * c))} \tag{1}$$

Where c is the balance factor, Eq. (2):

$$c = \frac{mean\ IP_{25}\ concentration}{mean\ phytoplankton\ biomarker\ concentration} \tag{2}$$

Brassicasterol has frequently been used as an open water biomarker; thus, it is often selected as the phytoplankton biomarker, when calculating the $PIP_{25}$ index ($P_BIP_{25}$). However, research suggests that in some environments brassicasterol may also be influenced by other sources, including sea-ice algae, freshwater and terrestrial sources (Huang and Meinschein, 1979; Volkman, 1986; Fahl and Stein, 2012; Belt et al., 2013, 2015). In these instances, dinosterol has been used as the phytoplankton biomarker for calculation of the $PIP_{25}$ index ($P_DIP_{25}$). In all three cores from Northeast Greenland, 90R, 109R and 134R, the concentration pattern of

brassicasterol is similar to dinosterol. Thus, the differences in the $PIP_{25}$ values are negligible (Supplement Fig. S3), and brassicasterol is deemed appropriate for use in the calculation. The mean brassicasterol concentrations were taken from the three cores combined, creating a regional average.

Correlation analysis was undertaken to assess the relationship between brassicasterol and $IP_{25}$ and dinosterol and $IP_{25}$ for each sediment core; the $R^2$ and p-values are presented for sites 90R, 109R and 134R respectively.

The ratio of two phytoplanktonic sterols: 1) epi-brassicasterol and 2) 24-methylenecholesterol (Bra/24-Me), details above, were calculated in the upper 8 cm of the cores 90R, 109R and 134R. This ratio, together with $IP_{25}$, can be used to investigate the degradation processes (Rontani et al., 2018). This is attributed to the different positions of double bonds in the alkyl chains of epi-brassicasterol and 24-methylenecholesterol, which makes enhanced reactivity more likely in epi-brassicasterol. Thus, an increase in the Bra/24-Me ratio has been used as an indicator of degradation (Rontani et al., 2018)

**3.7 Chronology**

Samples were taken at 0.5 cm intervals, freeze-dried and homogenised prior to analysis. All analyses were undertaken at the Leibniz Institute for Baltic Sea Research. Analyses (cf. Moros et al. 2017) of natural lead $^{210}$Pb and artificial $^{137}$Caesium ($^{137}$Cs) / $^{241}$Americeum ($^{241}$Am) radionuclides were carried out by gamma spectrometry with a Ge-detector (BE3830-7500SL-RDC-6-ULB) at IOW and processed with GENIE 2000 software (Canberra Industries Inc., USA). Analysis of mercury (Hg) was undertaken

using a DMA-80 Analyzer from MLS Company. This was calibrated against CRM (BCR) 142R certified reference material. Sample weights were 100 mg.

The combined event stratigraphic approach, applying radionuclide measurements and pollution records (Moros et al., 2017), has been successfully used in nearby areas (Perner et al., 2015, 2018, 2019). Due to low sedimentation rates, and the associated



uncertainty, linear interpolation was undertaken between three tie points: ~1900 (based on the increase in $^{210}$Pb and Hg which
indicates the increase in global atmospheric Hg pollution) and ~ 1954 (beginning of the atmospheric nuclear weapons testing
period = first trace of $^{137}$Cs and $^{241}$Am in downcore profiles), and the sediment surface (2017), the year of sampling. For comparison,
the upper part of core DA17-NG-ST08-092G was analysed for its Hg content.

### 3.8 Observational data

Monthly sea-ice concentration data was extracted from a historical sea-ice database (Walsh et al., 2019) for sites 90R, 109R and
134R respectively, from 1850 to 2017. The historical data is available as monthly concentrations on a 0.25 x 0.25-degree grid from
the National Snow and Ice Data Center (NSIDC). The site locations are therefore rounded to the nearest 0.25-degree latitude and
longitude for each site (Table 1) and extracted using the 'tidync' and 'dplyr' R packages alongside the averaged historical data for
the area covering all three site locations. Five-year running means were then extracted for comparison with the lower-resolution
biomarker data presented within this study. The September sea-ice cover is presented here, together with number of months with
>25% sea-ice cover.

Between 1850 and 1979 the historical observations are derived from ship observations, naval oceanographer compilations,
analyses by national ice services, with spatial/temporal interpolation based on regional climatology if no observational data was
available in the earliest sections of the record (Walsh et al., 2019). Between 1953 and 1973, the US Navy produced detailed regional
analyses of sea-ice concentration data which were spatially/temporally interpolated if the coverage was insufficient (Walsh et al.,
2019). After 1979, the dataset comprises sea-ice concentrations from satellite passive microwave data (Walsh et al., 2019).

### 3.9 Statistical analysis

Change-point analyses were undertaken on the observational sea-ice cover dataset (September sea-ice cover, 1850-2017; Walsh et
al., 2019) to detect the timing of changes occurring within the dataset, using the 'changepoint' software package from (Killick et
al., 2021). A series of models, which examine changes to the mean, variance and trend were applied, and the model with the best
fit was selected. This is calculated by examining the Akaike Information Criterion (AIC) and Bayesian Information Criterion (BIC)
estimates for each model. This analysis was undertaken on sites 90R, 109R and 134R separately.

The correlation between sterols (brassicasterol and dinosterol) and IP$_{25}$ was investigated. Biomarkers are normalised with
TOC. The same correlations with biomarkers normalised with sediment weight are presented in the Supplementary Material
(Supplement Fig. S1).

### 4   Results

#### 4.1 Chronology

The intact surfaces (0 cm) of all three Rumohr cores are assumed to correspond to the year of collection (2017), due to the
undisturbed sediment visible at the time of collection. $^{210}$Pb$_{unsupp.}$ surface values are 136, 146 and 180 Bq kg$^{-1}$ for 90R, 109R and



134R respectively; these values decrease exponentially downcore (Fig. 2). Hg concentration values range from 21 to 59 µg/kg, 39
to 62 µg/kg , and 41 to 63 µg/kg for cores 90R, 109R and 134R respectively (Fig. 2). The Hg values decrease downcore and reach
natural background level at 8.5 cm in 109R and at 7.25 in 134R. Core 90R is problematic as Hg values increase again from 7.25
cm downwards. Consequently, the Hg-based age constraint in this section of the core is uncertain. The ~1900 marker is thus placed
at 8.5 cm, 7.25 cm for cores 109R and 134R respectively. $^{241}$Am and $^{137}$Cs values first appear at 4.25 cm, 3.25 cm and 5.25 cm in
cores 90R, 109R and 134R respectively; this is marked as 1954 (Fig. 2).

Assuming a constant sedimentation rate between markers (2017, 1954 and 1900), the age model indicates that in the upper
part of the cores, the sedimentation rate is 0.05 cm yr$^{-1}$ (upper 3.25 cm), 0.05 cm yr$^{-1}$ (upper 3.25 cm) and 0.08 cm yr$^{-1}$ (upper 5.25
cm) in 134R, 109R and 90R respectively. Below these levels, the sedimentation rate is 0.05cm yr$^{-1}$ (134R) and 0.1 cm yr$^{-1}$ (109R).

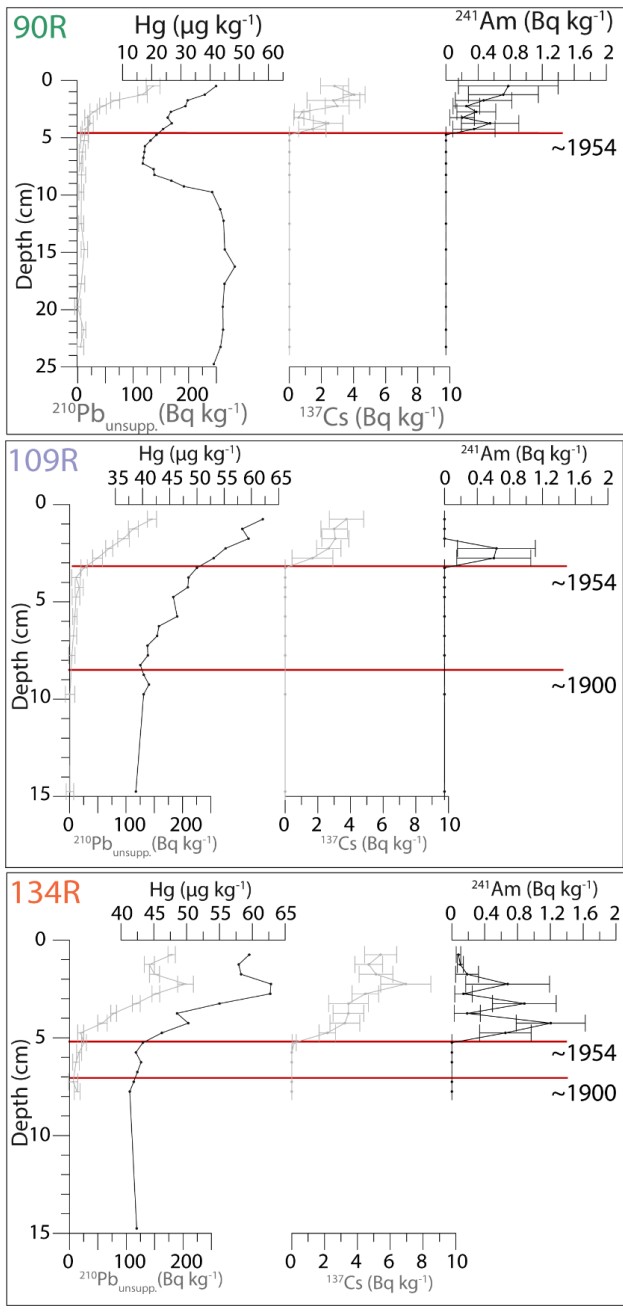

**Figure 2:** Downcore profiles of $^{210}Pb_{unsupp.}$, Hg, $^{137}Cs$, $^{241}Am$, including error ranges, for cores 90R (top), 109R (middle) and 134R (bottom);
colour of core name refers to the colour of the star for the site in Fig. 1. Approximate markers (1900 and 1954) are shown with red lines.






### 4.2 TOC, TN and $\delta^{13}C_{org}$

TOC values vary across the east-west transect (Fig. 3), with the lowest values located closest to the coast (90R; 0.62% average) and highest values in the mid (109R; 0.97% average) and outer trough (134R; 1.1% average) (Fig. 3). 90R has a TOC content of 0.92% in the lowermost part of the core, which decreases until 5.25 cm. From here, the TOC content gradually increases towards

the top of the core. 109R and 134R have lower TOC contents in the lower part of the core, but have the same, gradual increase towards the uppermost part of the core. The TN follows a similar pattern to the TOC in all cores with average values of 0.07%, 0.14% and 0.16% for 90R, 109R and 134R respectively (Fig. 3). $\delta^{13}C_{org}$ (‰ VPDB) fluctuates throughout all the cores (Fig. 3) with no obvious pattern.

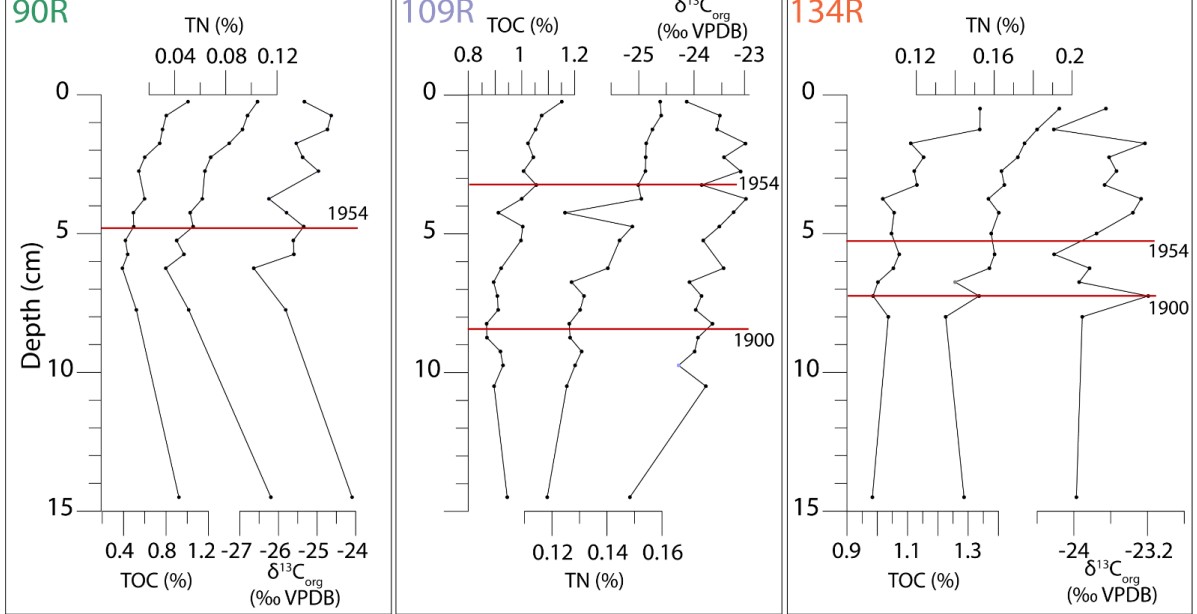

**Figure 3:** Total Organic Carbon (TOC), Total Nitrogen (TN) and $\delta^{13}C_{org}$ for 90R (left), 109R (middle) and 134R (right). The age markers (1900 and 1954) are shown for each core (red line).

### 4.3 Biomarkers

$IP_{25}$ concentrations vary between 90R, 109R and 134R, with average core values of 0.19, 0.11 and 0.06 µg g$^{-1}$ TOC respectively (Fig. 4). Patterns are similar for $IP_{25}$ concentrations normalised to TOC and dry sediment weight respectively (Fig. 4). $IP_{25}$ is

present in low concentrations in the lower parts of 90R and 109R, and completely absent below 6 cm depth in 134R. Concentrations begin to increase at 7.75 cm in 90R and at 9.5 cm in 109R, and increase gradually towards the upper most part of the cores, with a slight decrease in the upper 1 cm. There is a peak in $IP_{25}$ at 5.75 cm in 134R, and concentrations increase from 4 cm towards the top of the core, again with a slight decrease in the uppermost section (Fig. 4).

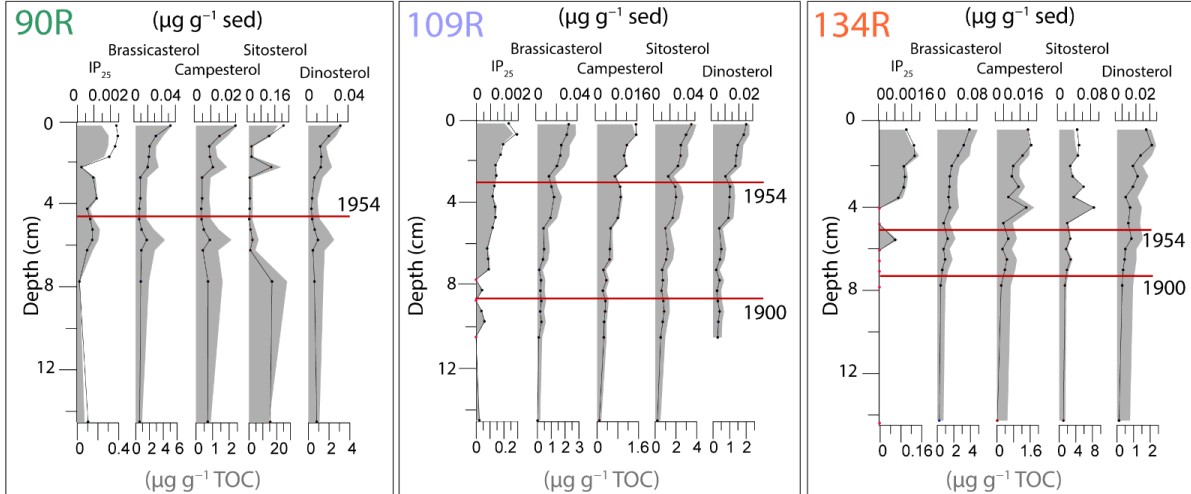

**Figure 4:** Concentrations of biomarkers, IP$_{25}$ and sterols (brassicasterol**,** campesterol, sitosterol**,** and dinosterol for 90R (left), 109R (middle) and 134 (right) shown on depth. Red dots on the IP$_{25}$ curve indicate samples where IP$_{25}$ was present but was too low to quantify. Concentrations are normalized using the dry sediment weight (dark-coloured lines) and TOC (grey shaded area). The age markers (1900 and 1954) are shown for each core (red line).

In core 90R, concentrations of brassicasterol, campesterol, and dinosterol are relatively low before concentrations increase from

2.25 cm towards the uppermost part of the core (Fig. 4). There is a noticeable peak in the same three sterols at 5.75 cm. Sitosterol concentrations are slightly higher in the lower two samples of the core, with low concentrations between 6.25 cm and a slight increase from 1.25 cm towards the top of the core (Fig. 4). The concentration of all sterols in 109R are low but increase steadily upwards through the core, with the highest concentrations in the surface sediments. The pattern of sterol concentrations in core 134R are similar to those of 109R, with a gradual increase towards the top of the core. The exception is sitosterol, which fluctuates

between 4 and 0 cm, with no clear increase. Concentrations of sterols normalised using TOC and dry sediment weight show a similar pattern for all cores (Fig. 4).

Correlation plots (Fig. 5) show that there is a significant (p<0.05) positive correlation between brassicasterol and IP$_{25}$ in cores 90R, 109R and 134R with R values of 0.62 0.90 and 0.78 respectively. The exceptions are samples where the IP$_{25}$ concentrations were too low to be quantified. Dinosterol and IP$_{25}$ are also significantly correlated (p<0.05) for all three cores, with R values of

0.56, 0.88 and 0.82 for 90R, 109R and 134R respectively. 134R has the lowest IP$_{25}$ concentrations, with the values in 109R and 134R being more similar (Fig. 5).



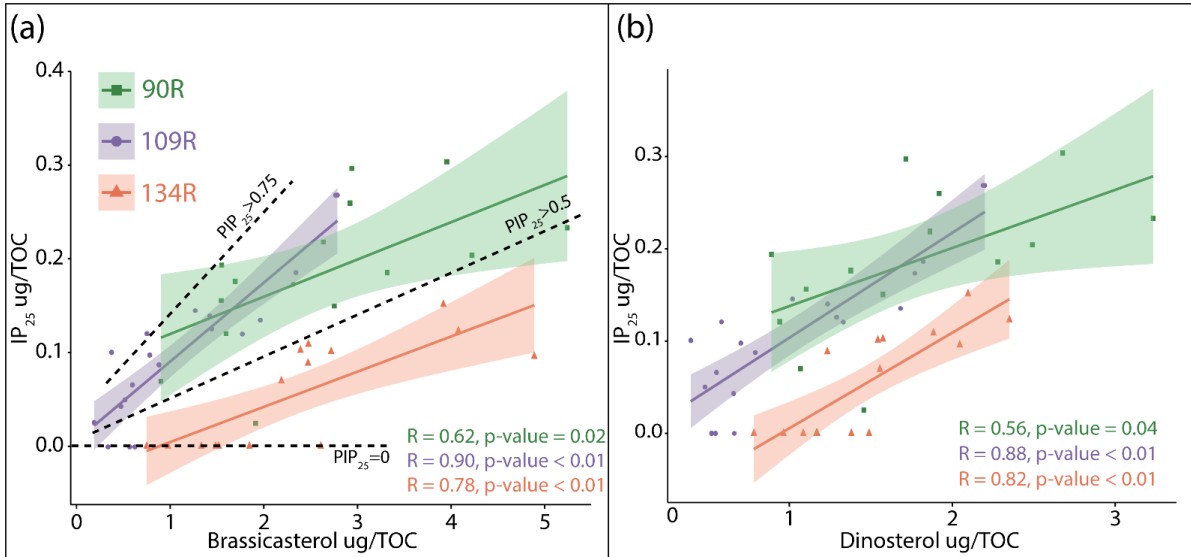

**Figure 5:** Correlations of brassicasterol and IP$_{25}$ (a) and dinosterol and IP$_{25}$ (b) for all sediment cores: 90R (green), 109R (purple) and 134R (orange). IP$_{25}$ in normalised with TOC. The R values are listed for each core (bottom right) together with the p values. Black dashed lines are used to separate samples based on PIP$_{25}$ values (0.75, >0.5 and = 0).

### 4.4 Ratio of epi-brassicasterol and 24-methylenecholesterol (Bra/24-Me)

In some samples, 24-methylenecholesterol could not be quantified; consequently, the ratio of Bra/24-Me is not calculated in these instances. The ratio of Bra/24-Me (Fig. 6) is relatively stable in the lower section of 90R (7.75 to 2.75 cm), before peak values at 2.25 cm, which corresponds to a dip in IP$_{25}$ concentrations. From here values remain relatively low and stable (upper 1.75 cm), as IP$_{25}$ values increase simultaneously. The ratio of Bra/24-Me gradually decreases, but with some fluctuations, towards the upper section of core 109R, this corresponds to a general gradual increase in IP$_{25}$ concentrations (Fig. 6). There are fluctuations in the Bra/24-Me ratio between 5.75 cm and 3.25 cm in core 134R, before values decrease and remain relatively low towards the sediment surface (Fig. 6).



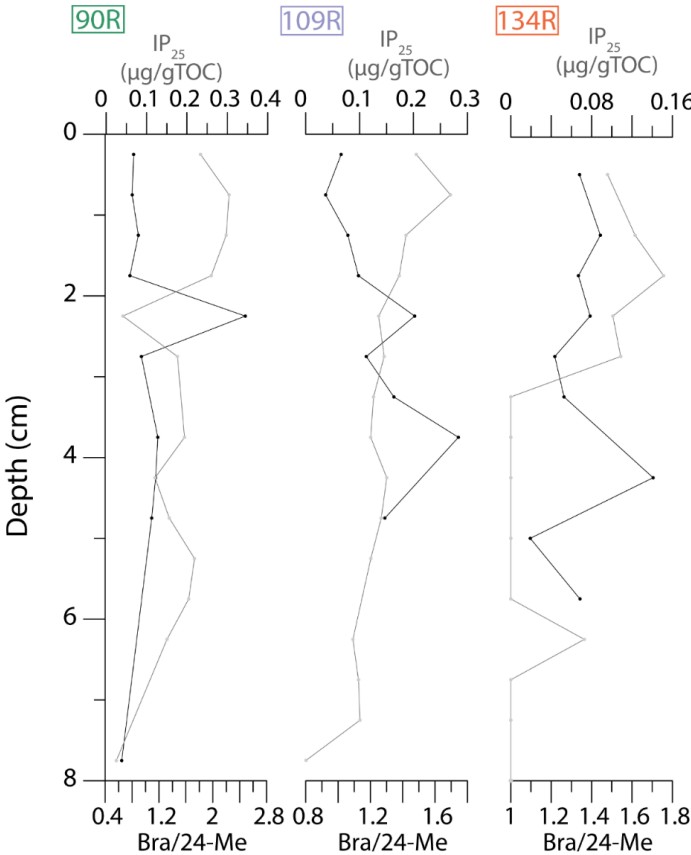

**Figure 6:** Downcore plots (upper 8 cm) of IP$_{25}$ concentrations (grey) and epi-brassicasterol/24-methylenecholesterol (Bra/24-Me) ratio (black) for 90R (left), 109R (middle) and 134R (right).

### 4.5 Sea-ice cover observational record

The sea-ice record between 1850 and 2017 CE, based on satellite data and historical observations, provides an overview of sea-ice conditions at sites 90R, 109R and 134R (Fig. 7(Walsh et al., 2019, 2017). Throughout the entire records, site 134R is consistently characterised by the lowest September sea-ice cover (56% on average), while 90R has the most extensive sea-ice cover (87% on average), with the sea-ice cover at 109R of intermediate values (76% on average).

Change-point analysis, undertaken on the sea-ice cover data at sites 90R, 109R and 134R, reveals key changes to sea-ice cover throughout the time period (Fig. 7). The main change is identified using this analysis and visual interpretation of the September sea-ice cover data and the number of years when the monthly sea-ice cover is > 25%. This occurs at 1971 (sites 90R and 109R) and 1984 (134R). Between 1900 and 1971 the average September sea-ice cover at sites 90R and 109R is 91% and 82% respectively. Between 1971 and 2017 the average sea-ice cover is 87% and 64% at the same sites. Sea-ice cover at site 134R is 63% between 1850 and 1984 and 36% from 1984 onwards.

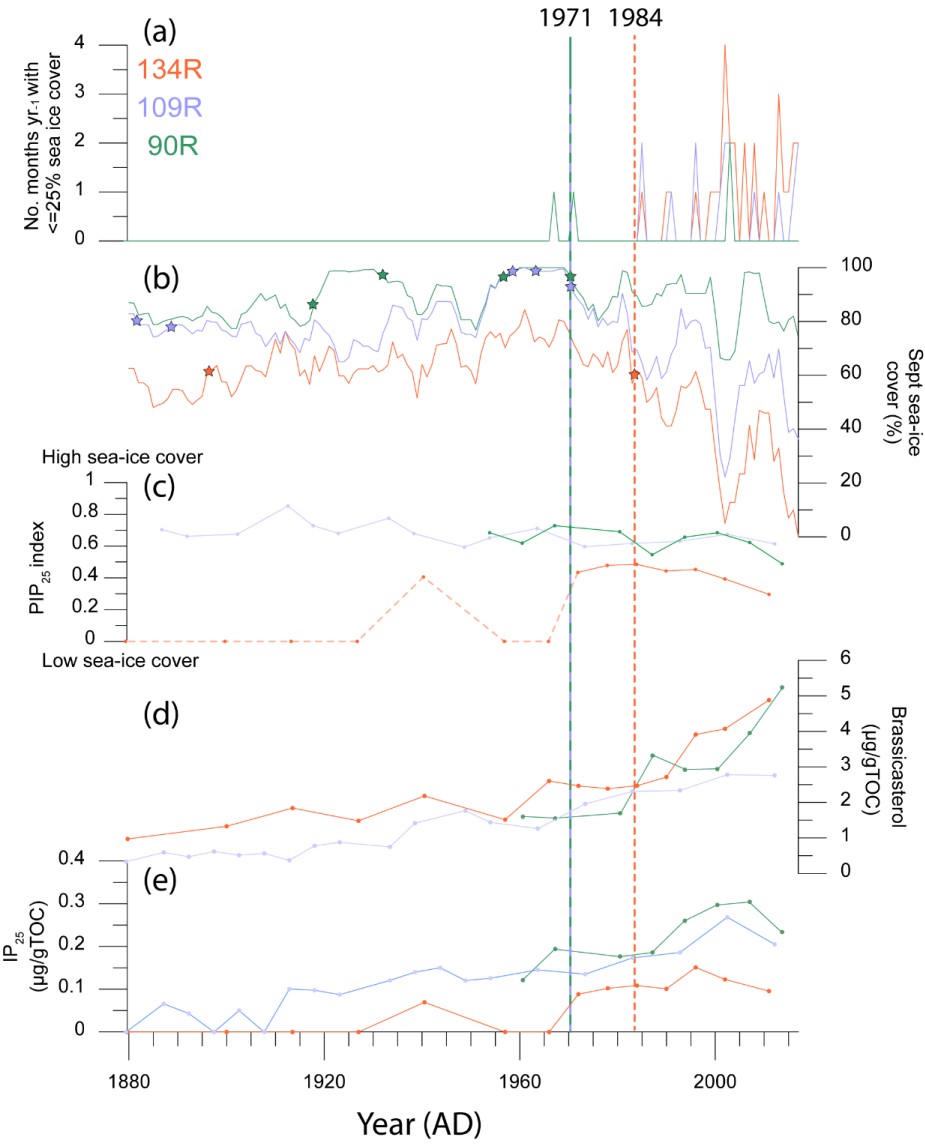

**Figure 7:** Sea-ice data from instrumental and historical records (a-b) and biomarker data (c-e) for cores 90R (green), 109R (purple) and 134R (orange). (a) The number of months with sea-ice cover <= 25% for all three core sites. (b) the average September sea-ice cover for all three sites, with a 5-year running mean applied for each site. All instrumental and historical data (a and b) is extracted from Walsh et al., (2019). (c) The PIP$_{25}$ index values for all three cores. The early part of the PIP$_{25}$ record for 134R is marked with a dotted line, as IP$_{25}$ was absent or too low to be detected (d) brassicasterol concentrations, and (e) IP$_{25}$ concentrations. Biomarkers are normalised using Total Organic Carbon (TOC). Due to uncertainty in the age-model, biomarker data for core 90R is not included pre-1954 (c-e). Changes in sea-ice cover, identified using change-point analysis marked with stars for cores 90R (green star), 109R (purple star) and 134R (orange star). Key changes in sea-ice cover, identified using the change-point analyses and visual interpretation of sea-ice data (a and b) are marked with a dashed vertical line and labelled for sites 90R (green), 109R (purple) and 134R (orange).



## 5   Discussion

### 5.1  Record uncertainty

Prior to the discussion of biomarker-based sea-ice reconstructions of the Northeast Greenland continental margin/shelf area, it is necessary to assess uncertainty within these records; specifically, issues relating to age constraint, bioturbation and reworking of material, and the degradation of biomarkers in the redox sedimentary boundary layer.

### 5.1.1     Chronology

Sedimentation rates from the upper sediments of nearby cores, that are located in, or in close proximity to our transect in the Belgica Trough are: 0.004 cm yr$^{-1}$ (Lloyd et al., 2023), 0.05 cm yr$^{-1}$ (Syring et al., 2020b), 0.009 yr$^{-1}$ (Davies et al., 2022). Sedimentation rates in cores 90R, 109R and 134R are within the same range as these cores, but generally higher: ranging from 0.04-0.08, 0.05-0.1 and around 0.05 cm yr$^{-1}$ respectively. The relatively slow sedimentation rates at these sites means that each sediment sample encapsulates more than one year of sea-ice cover: in the upper sediments, the sampling resolution of 0.5 cm

corresponds to between 6 and 10 years per sample. Thus, the changes in biomarker and sterol data will not directly reflect high-resolution patterns captured in the instrumental data.

Whilst CT scans from nearby gravity core DA17-ST0-092G reveal bioturbation in the upper sediments (e.g. Davies et al., 2022), the steady decline in $^{210}$Pb$_{unsupp.}$ $^{137}$Cs and Hg in sediment cores 109R, and 134R suggests that reworking is not affecting the chronology to a large extent. However, the reversal of the otherwise steady downwards decline in Hg concentrations in core 90R

at around 7.25 cm, where concentrations rise again. This is potentially caused by reworking or input of terrigenous material which could also explain the peaks in some sterol biomarkers, including terrestrially sourced sterols (sitosterol and campesterol). At this depth, the fine grained sediments and water content decrease, as the sand content increases (Supplement Fig. S2). In contrast, the $\delta^{13}C_{org}$ increases at this depth, suggesting that there is less terrestrial material. Furthermore, examination of the XRF element ratios, Ca/Fe and Ca/Sr, which are often used to differentiate between marine and terrestrial Ca, together with greyscale, reveal no clear

changes at this depth (Supplementary Material).

Uncertainty within the age models of these cores also stems from the fact that each is based on only two (90R; 2017 and 1954) or three tie-points (109R, 134R; 2017, 1954 and 1900). This means that a direct comparison with instrumental data may represent different time periods. Thus, only the general trends are considered.

### 5.1.2     Degradation of biomarkers

The relative stability of IP$_{25}$, compared to other phytoplanktonic lipids, makes it an attractive proxy for sea-ice reconstructions on longer timescales (Stein et al., 2016; Clotten et al., 2018, 2019; Rahaman et al., 2020; Detlef et al., 2021, 2023), as it most likely reflects sea-ice signals rather than degradation activity (Rontani et al., 2011, 2014). However, near-surface sediment studies often span the redox sedimentary boundary, making diagenetic transformations in these sections more likely (Rontani et al., 2018). Unfortunately, in the Arctic, the core depth level of biomarkers preserved in the upper sediments frequently coincide with dramatic

either anthropogenically driven changes in sea-ice cover, visible in instrumental data (Meier and Stroeve, 2022), or naturally driven



climate-related changes in sea-ice cover (Xiao et al., 2015a). As a result, disentangling diagenetic signals and those driven by anthropogenically and/or naturally driven changes in sea-ice cover is of upmost importance in such studies.

The decline in $IP_{25}$ down-core in a short record from the Chukchi-Alaskan margin was attributed, in part, to diagenesis (Polyak et al., 2016). However, in many areas the general agreement between short core biomarker records and historical sea-ice data is good, with some sites even showing increases of $IP_{25}$ with depth (Andrews et al., 2009; Vare et al., 2010; Alonso-García et al., 2013; Cormier et al., 2016), suggesting that preservation here is not an issue. Additionally, whilst $IP_{25}$ oxidation products were found from cores in the Barrow Strait and Amundsen Gulf in Arctic Canada, indicating some degradation, a negative trend in $IP_{25}$ concentrations was not visible, implying a dominant sea-ice signal rather than diagenetic transformation (Rontani et al., 2018). This evidence indicates that whilst caution should be applied in short-core studies, near-surface changes in $IP_{25}$ concentrations do, most likely, reflect sea-ice cover in many instances.

Whilst the general trend of $IP_{25}$ concentrations in our cores from Northeast Greenland (90R, 109R, and 134R) is declining downcore from the surface sediments, there is some evidence that suggests sea-ice cover is driving this pattern, rather than diagenesis. Firstly, there is an increase in $IP_{25}$ concentrations from the surface sample downcore (in the upper few centimetres) in core 134R and 90R, contrary to a purely degradation-driven signal. Furthermore, a comparison with the $PIP_{25}$ index, calculated with the $IP_{25}$ concentration data, shows good agreement with instrumental data. However, to ascertain these conclusions, we examine other degradation markers, namely the ratio of epi-brassicasterol and 24-methylenecholesterol.

An increase in the ratio of epi-brassicasterol and 24-methylenecholesterol (Bra/24-Me), together with a decreasing $IP_{25}$ concentrations downcore, has previously been used to identify autoxidation processes, which cause the degradation of $IP_{25}$ in near-surface sediments (Rontani et al., 2018). From the sediment surface, there is a general decline in the concentration of $IP_{25}$ downcore in 90R, 109R, and 134R. At the same time, the ratio of Bra/24-Me is generally stable in 90R (0.25 to 7.75 cm), apart from a peak at 2.25 cm, and 134R (between 0.5 and 3.25 cm). In contrast, the Bra/24-Me ratio generally increases downcore in 109R (0.25 to 3.75 cm; Fig. 6). This suggests that degradation processes may be contributing to the decline in $IP_{25}$ downcore in 109R, yet it is unlikely in 90R and 134R. Collectively, we argue that this evidence indicates that sea-ice cover is the primary driver of changes in $IP_{25}$ concentrations in cores 90R and 134R from Northeast Greenland. However, oxidation processes may be contributing to the decline in $IP_{25}$ from the sediment surface in core 109R.

## 5.2 Sea-ice cover

### 5.2.1 Northwest to Southeast Transect

Surface sample concentrations of $IP_{25}$ are 0.233, 0.205, and 0.096 μg g$^{-1}$ TOC (or 0.0024, 0.002 and 0.0013 μg g$^{-1}$ sediment) for 90R, 109R, and 134R, respectively. This reflects the current sea-ice conditions in the region, with more extensive sea ice located at the inner shelf (90R) and reduced ice cover at the outer shelf (134R). Overall, these much lower than seen in previous surface sample measurements, using the same internal standards, from the Belgica Trough, which vary from 3.31 to 23.44 μg g$^{-1}$ TOC (or 0.016 to 0.043 μg g$^{-1}$ sediment) (Kolling et al., 2020). We argue that this may be attributed to the storage of sediments (e.g. frozen vs cooled), where biomarkers may have degraded over time in our sediment samples stored in the refrigerator. It is also possible



that the very upper sediments were disturbed or missing, despite an intact surface being visible at the time of collection. However,

the pattern is still similar to our sites, with lower values generally located on the outer shelf and higher values at the coastal sites (Kolling et al., 2020). Concentrations of brassicasterol in the surface samples from this study are 5.23, 2.77 and 4.89 μg g$^{-1}$ TOC (or are 0.053, 0.032 and 0.065 μg g$^{-1}$ sediment) for 90R, 109R, and 134R. Brassicasterol concentrations from samples located nearby are much higher, with values ranging from 10.81 to 68.88 μg g$^{-1}$ TOC (or 0.094 to 0.34 μg g$^{-1}$ sediment) and the higher values generally located on the outer shelf (Kolling et al., 2020). Again, the differences may be attributed to the storage of sediments

after collection.

     The PIP$_{25}$ values for site 134R are lower than 90R and 109R throughout the entire study period. This broadly aligns with observational sea-ice records, which indicate less extensive sea-ice cover at this site from 1880 to 2017 compared to more coastal locations. The PIP$_{25}$ values for 90R and 109R are, on the other hand, more similar, despite observational data showing that sea-ice cover is generally lower at 109R.

Whilst the PIP$_{25}$ index provides a useful way to assess see-ice cover, it is important to consider individual biomarkers as well; thus, we examine the biplots of IP$_{25}$ and sterols (Fig. 5). The positive correlations between IP$_{25}$ and brassicasterol, and IP$_{25}$ and dinosterol at all three sites (Fig. 5) can best be explained by the fact that under more extreme sea-ice conditions, both biomarkers show low values but with decreasing sea-ice, indicating more open-water and ice-edge conditions. Furthermore, both biomarker concentrations increase as open-water production and sea-ice-algae production increase, implying a decrease in sea-ice cover.

Early in the 90R record (5.75 cm core depth; sometime before 1954 as the age constraint is uncertain here), there is a peak in brassicasterol, dinosterol and campesterol, together with a slightly higher IP$_{25}$ concentration. However, uncertainty in the age constraint in this section of the core should be considered, this is discussed in detail in section 5.1.1. The high sterol concentrations in the lower part of 90R suggests that the inner Belgica Trough may have been characterised by open water conditions with some seasonal sea-ice cover. Such elevated concentrations of IP$_{25}$ and the open-water phytoplankton biomarkers is typical for polynya-

situations (see Stein et al., 2017; Syring et al., 2020a). In contrast, at site 109R, all sterol and IP$_{25}$ concentrations are lower during this period, resulting in a higher PIP$_{25}$ index value. The evidence that IP$_{25}$ and the open-water phytoplankton biomarkers are very low or absent, suggests more extensive to permanent sea-ice cover in this area (cf., Müller et al., 2011). Considering these results, we suggest the presence of a regional polynya-like feature in the inner most Belgica Trough (90R), sometime before 1954. Such findings are not visible in the historical data during this period. As this data comes from historical records and uses spatial analogue

gap filling (Walsh et al., 2019), it is unlikely that such a feature would be captured this early within the observational record.

     Well defined polynyas, or areas of open water, are common features in the transitional ice zone between the land-fast sea-ice and the pack ice (Pedersen et al., 2010). Several polynyas currently form along the coast of Northeast Greenland: the NEW polynya, Sirius Water polynya, the Scoresby Sund Water polynya, Store Koldewey polynya, and a polynya located south of Île-de-France, (Sørensen, 2012). These polynyas are formed by the presence of land-fast sea-ice, together with ocean currents and katabatic winds

(Schneider and Budéus, 1994; Pedersen et al., 2010). Biomarker evidence from 90R, together with existing polynyas along the East coast of Greenland, suggests that a polynya-like feature could have been a plausible feature forming to the east of Norske Øer as also has been proposed in a Holocene sea-ice biomarker record obtained from near-by Core PS93/025 (Syring et al., 2020a; for core location see Fig. 1).





### 5.2.2    Temporal changes in sea-ice cover

The presence of IP$_{25}$ in most of the samples in 90R and 109R suggests that seasonal sea ice has been present for the last ~120 years in the coastal and mid part of the Belgica Trough. In the early part of the record, the concentration of IP$_{25}$ and brassicasterol is generally stable, with some small fluctuations. This aligns with the observational record, which shows a similar pattern.

IP$_{25}$ is absent, or only present in very low concentrations, in the lower part of 134R. Together with the presence of brassicasterol, this results in low PIP$_{25}$ values and indicates the absence of seasonal sea ice. This differs from the observational and 420    instrumental data for sea-ice concentrations in this region, with an average of 59% cover, during the same time-period (1880-1920). However, it should also be noted that the absence of IP$_{25}$ may be a result of other factors: 1) lack of or only few diatoms in the sea-ice, 2) removal of algae in the water column by grazing 3) degradation in the water column and sediments (Belt, 2018). Thus, one of the aforementioned reasons may be the reason behind the discrepancy between instrumental data and our reconstructions in the earlier parts of the record in core 134R.

The general increase in IP$_{25}$ and brassicasterol at sites 109R and 90R from ~1970 suggests a shift towards more seasonal sea-ice cover and open water conditions. This aligns with results from the change-point analysis of observational data; 1971 represents a significant change in sea-ice cover at site 90R with values generally declining from 0.7 (1967) to 0.5 (2014), and a noticeably steeper decline from 1994. Change-point analysis for sea-ice cover at site 134R reveals a slightly later change in 1984. From ~1984, the decline in sea-ice cover reconstructed using the PIP$_{25}$ index for 134R aligns with the patterns in instrumental records; 430    value decline from 0.65 (1984) to 0.4 (2011). The PIP$_{25}$ values during this period are more stable at 109R, not reflecting the observational record.

The declining sea-ice concentrations since the 1970-1980 period, found in our data from sites 90R, 109R and 134R is likely linked to the ongoing Atlantification, characterised as significant warming (1 °C) and shoaling of Atlantic Water. This has been observed across the Northeast Greenland continental shelf in recent decades (Gjelstrup et al., 2022). In the Belgica Trough, CTD 435    records indicate a warming of 0.5 °C of Atlantic Waters between 2000 and 2016, relative to 1979-1999 (Schaffer et al., 2017). This has implications for sea-ice cover, including the NØIB, via enhanced melting from below (Sneed and Hamilton, 2016; Gjelstrup et al., 2022). Observations suggest that there was more summer sea ice melt in the 2000s compared to the 1990s, leading to the freshening of surface waters (Gjelstrup et al., 2022).

### 6    Conclusions

High resolution biomarker data (IP$_{25}$, PIP$_{25}$, brassicasterol and Bra/24-Me) from three marine sediment cores collected from a transect, spanning the inner to outer Belgica Trough on the Northeast Greenland continental shelf, together with instrumental and historical sea-ice cover data provides the following conclusions:

1. Uncertainty in the chronological constraint of sediment cores 90R, 109R and 134R, based on $^{210}$Pb, $^{137}$Cs and Hg concentration data, stems from the limited number of age-markers (2-3 points). This makes a direct and statistical 445    comparison with instrumental data difficult; rather, the general trends are assessed. For cores 109R and 134R, the age model is reliable until 1900, where the earliest age marker is found ($^{210}$Pb reaches base level). The age constraint prior to



1954 in core 90R is uncertain, due to the increasing concentration of Hg downcore, this is attributed to material reworking or the presence of a turbidite.

2. Degradation of biomarkers is often an issue in the redox sedimentary boundary layer. However, biomarker evidence, specifically the decline in $IP_{25}$ in the surface sediments and peaks in $IP_{25}$ in lower parts of record in 134R suggests it is less affecting the results here. Furthermore, evidence from the Bra/24-Me ratio suggests degradation has not been affecting the biomarkers in the upper sediments in core 90R and 134R; it may have affected 109R to some extent.

3. Historical and satellite data of sea-ice cover shows that site 134R (outer shelf) has been characterised by the lowest September sea-ice cover, while site 90R (coast proximal) has had the highest sea-ice cover throughout the entire period covered by these data (1880-2017). In accordance with this, site 90R has been characterised by the highest $IP_{25}$ values, and site 134R with the lowest values throughout our record. This again results in the lowest $PIP_{25}$ values (least sea ice) at site 134R, with higher values at 90R and 109R (most sea ice) respectively, thus aligning with the observational record from this region throughout this entire period.

4. A decline in sea-ice cover visible in the instrumental record, and identified using change-point analysis, begins in 1971 at sites 109R and 90R. This aligns with increasing brassicasterol and $IP_{25}$ concentrations in these sediment cores, suggesting a shift to more seasonal sea-ice cover and open water conditions after ca. 1970. This change occurs later in at the outer shelf site of core 134R (1984); this aligns with the sharp increase in brassicasterol and decline in the $PIP_{25}$ index at this time. This change is likely linked to the ongoing Atlantification and warming of surface waters that have been observed in recent decades.

*Author contributions.* JD, MSS and CP designed this experiment, with help from RS, KF and HD. MSS led the NorthGreen17 Expedition. JD carried out formal analysis for sea-ice biomarkers with help from KF. Funding was acquired by MSS and CP. ACC undertook statistical analysis. MM undertook the analysis for the chronological constraint of all sediment cores. JD prepared the manuscript with valuable contributions from all co-authors.

*Competing interests.* The authors declare that they have no conflict of interest.

*Acknowledgements.* We thank the captain, crew and scientists onboard the NorthGreen2017 expedition on RV Dana. We thank Anne de Vernal for help acquiring funding for the NorthGreen 2017 Expedition. Particular thanks also go to Walter Luttmer for his expertise and assistance in the biomarker laboratory at AWI. We acknowledge the use of imagery from the NASA Worldview application (https://worldview.earthdata.nasa.gov/), part of the NASA Earth Observing System Data and Information System (EOSDIS).

*Financial support.* The NorthGreen2017 expedition was funded by the Danish Centre for Marine Research and the Natural Science and Engineering Research Council of Canada. The research was funded by the Independent Research Fund Denmark (grants no.



7014-00113B (G-Ice) and 0135-00165B (GreenShelf) to MSS), and by the iClimate Centre of Aarhus University, with additional funding from the European Union's Horizon 2020 research and innovation program under Grant Agreement No. 869383 (ECOTIP).

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
