# Peer review of "years of sea-ice conditions on the Northeast Greenland continental shelf: a biomarker and observational record comparison"

_EGUsphere, 2023_

## Author Response (AR1)

Dr. Sebastian Gerland
Editor, The Cryosphere

Please find responses to both reviewer's comments in the text below for the manuscript entitled: '*120 years of sea-ice conditions on the Northeast Greenland continental shelf: a biomarker and observational record comparison*'.

We would like to thank the two reviewers and the editorial team for these comments and have outlined our responses in red.

Kind regards,

Joanna Davies

**Reviewer 1**

The authors present a comprehensive biomarker data set from three NE Greenland shelf records together with instrumental/observational data for sea ice fluctuations over the last century. A well-balanced and interesting manuscript with plenty of new datasets certainly of interest for the readership of EGUsphere. I would like to highlight a few critical points that might be addressed before the manuscript can be accepted for publication:

We are happy that the reviewer believes that this dataset is of relevance and interest to the readership of the EGUsphere.

First, the authors have access to bulk organic information including TOC, TN, and d13Corg While the bulk organics give you a comprehensive overview on the organic matter sources, the biomarkers cover only a tiny fraction of it. You could use the data better to inform the readership of dominant organic matter source in the records. You may even consider a rough semi-quantification of marine and terrestrial organic matter and use is more actively for your interpretation. The d13Corg data vary between -23 and -27 permille implying quite a bit of variation in terms of terrestrial organic matter supply to your shelf system.

We understand the idea behind this comment, however previous studies from Northeast Greenland have shown that the link between $\delta^{13}C_{org}$ from land and marine environments is complicated to decipher in this area (e.g. Andreasen et al., 2023. Boreas). We agree that further studies are needed to determine organic matter sources. However, this is not the focus of our own study, which aims to understand sea-ice conditions instead; thus, we have decided to not include this aspect in the present paper. However, we have justified why we have not used this dataset to determine the source of organic matter this in the text (lines 148-150).

The authors (desperately) try to argue that the near-surficial deposits are less influenced by bio-degradation compared to the climate signal preserved within the biomarker records. Rontani et al. (2018) is often referred while only the ration of epi-brassicasterol and 24-methylenecholesterol is shown. Why don't you analyse the autoxidation products of IP25 in some of your samples? You have the co-authors to do this experiment. It would strengthen your dataset immensely and avoid mis-interpretation of your data.

Thanks for this suggestion, we agree that providing more evidence related to degradation of the biomarkers would improve our arguments relating to a climatically driven signal. As such

we have analysed and quantified the autoxidation products of $IP_{25}$ (a) 2,6,10,14-tetramethyl-9-(3-methylpent-4-enyl)-pentadecan-2-ol and (b) 2,6,10,14-tetramethyl-7-(3-methylpent-4-enyl)-pentadecan-2-ol) as suggested. These results are presented as a separate figure in the manuscript, plotted with the $IP_{25}$ concentrations (Fig. 7) and described in a new section in the results (Section 3.5). As per the Bra/Me-24 data, our results from the autoxidation products suggest that degradation is not affecting the $IP_{25}$ signal in two of the sediment cores (134R and 90R). This is presented in the discussion (Lines 396-397).

Clearly, from the discussion, core 109R is affected by biodegradable products. (from the Bra/24-Me) ratio. You may run some of your fractions again for potential prevalence of autoxidation products of IP25 as well.

As outlined in our previous response, we have run some of our fractions in all cores to identify the autoxidation products of $IP_{25}$; this is presented in the manuscript in the results and discussion.

Also, the gradual decline in brassicasterol concentration in all records could be interpreted as a result of diagenesis.

We agree with the reviewer that this is an important point, so have added this to the discussion about diagenesis (Line 380).

Perhaps the application of PIP25 is here rather speculative and taken the uncertainties of biodegradation into account, I would suggest to leave it out. You have a visually good correlation with declining sea cover from your observational data set. According to Rontani et al. (2018) this is your strongest argument against significant bio-degradational control.

We agree that the $PIP_{25}$ index results are speculative, however we believe that they provide useful information when combined with discussion of the individual biomarker results. We believe that these values are useful for comparison amongst sites, e.g. to show lowest sea ice cover at site 134R throughout the entire study period. As such, we have used the $PIP_{25}$ index for a small part of the discussion still, however we use the individual biomarkers for the majority of the discussion.

Minor comments

- You mention X-ray fluorescence scanning and grain size analysis in the methods, but you hardly use these data for your interpretation. Consider showing the data actively or omit. X-ray fluorescence data can also provide you with information on diagenesis (redox boundaries).

As XRF and grain size analysis was only used to correlate the gravity (DA17-NG-ST08-092G) and Rumohr core (DA17-NG-ST08-090R) we believe that it is important to include in the methods. However, we don't present this data for the other cores so have not used it to identify potential redox boundaries. This is now outlined more clearly in the methods section of the manuscript (Lines 129-130, Line 134).

- You may provide more details to your bulk analysis including d13Corg measurements, uncertainties, errors, standards etc.

We thank the reviewer for this comment, this has been added to the methodology section 3.5.

**Reviewer 2 – Florence Fetterer**

I read the article from the standpoint of someone familiar with sea ice data sets, but not at all familiar with biomarkers for sea ice. The authors state that the outcomes "support the reliability of biomarkers for sea-ice reconstruction in this region". They present biomarker evidence that a "polynya-like feature" may have been forming in the westernmost Belgica Trough sometime before mid-century. This is an interesting finding, and it illustrates how these proxy data can be used when other sources fall short. The work contributes to understanding the history of sea ice area and extent off NE Greenland, and this, as the authors note, contributes to understanding the dynamics of two important glaciers that are buttressed by sea ice.

We are happy that the reviewer believes this work is of importance in understanding the history of sea ice in Northeast Greenland

The historical sea ice data set that biomarker data are compared with (Walsh et al., 2019) has large uncertainties, but I think it is used appropriately here. The authors are not correlating sea ice concentration percent values from the historical data with bioindicator values, rather, they are considering only relative amounts of sea ice cover and adding strength to their interpretation with change-point analysis.

Overall, the paper is well constructed and well written. Terminology needs clarifying in places, if only to help cross-discipline readers. I've called out those places below.

Specific comments

In the Introduction, it would be helpful to give a few descriptive words when IP25 and PIP25 first appear. This would be a kindness to those of us who know nothing of biological proxies for sea ice but want to learn how they can be used along with the satellite and other observational records we're familiar with.

We thank the reviewer for this comment and have defined these descriptive words more clearly in the text (Line 36, Line 40).

Ln 28. Please define sea-ice cover here. "Sea-ice cover" can be confused with sea ice extent, when what I think you mean is sea ice concentration or area. (NSIDC has a short piece on "What is the difference between sea ice area and extent?", by the way.) "Sea-ice cover" is fine to use if how it is being used is made clear.

This is a good point. When we are referring to the observational record, we are referring to sea-ice concentration data. Therefore, we have changed the terminology when referring to the observational record. However, as sea-ice biomarkers (specifically the $PIP_{25}$ index) produce a semi-quantitative estimate of sea ice we believe that sea-ice concentration is not the best term to use for this. Sea-ice cover or sea-ice conditions are widely used to describe sea-ice biomarker data. Therefore, when we refer to sea-ice biomarker data we have used these terms accordingly.

Beginning line 59 is: "Northeast Greenland is an area characterised by several sea ice types and features; it is thus a region of interest to understand the impact of climate changes on sea-ice extent. These features include land-fast sea ice (hereafter 'fast ice'), seasonal sea ice and the Northeast Water (NEW) polynya."

I'd like to better understand how the authors are using "seasonal sea ice". Usually, the term refers to broad expanses of ice that form in the winter months but are ice free in the summer; that is, the region between the ice edge in winter and the ice edge in summer. The Belgica Bank area has not typically experienced this type of seasonal ice. Looking quickly at the monthly extents in passive microwave satellite data, August and September of 2021 are the only times I see the sea ice extent retreat north of the Belgica Bank, although ice retreats well to the west in 2017.

In typical usage, "seasonal" means ice is there in the winter but not in the summer. But ice off the NE coast of Greenland is always there (except, notably, in summer 2021). If the authors are thinking of seasonal ice as ice cover interrupted over time by polynyas, or just by variable areas of open water between pack ice floes moving south as shown in Figure 1, I recommend using a different term.

We agree with the reviewer on this point, and this terminology has been changed throughout the manuscript.

Ln 57. It's not necessary to include this but I want to note that the Divine and Dick data are available at NSIDC:

D.V. Divine, C. Dick. March through August Ice Edge Positions in the Nordic Seas, 1750-2002, Version 1 NSIDC: National Snow and Ice Data Center, Boulder, Colorado USA (2007), 10.7265/N59884X1

We thank the reviewer for noting this publication

Around Line 90, suggest you reference Fig 1(b) in the same sentence that first mentions the two marine-terminating glaciers. Why is one glacier labeled NG when first introduced, and labeled "79NG" on the figure and in the text in later mentions?

This was a mistake and has been changed in the manuscript accordingly to 79NG (now Line 92).

Ln 126-129 Curious as to why this X-ray fluorescence step was not carried out for the other cores. Consider adding a sentence as to why. Also for the grain size analysis step. Were these steps done just to check the match between Rumohr core and gravity core results for roughly the same location?

We thank the reviewer for this comment. We have added information to explain that XRF and grain size analysis was carried out on core 90R for the purpose of correlation with 92G. This was undertaken for the purpose of age constraint and was not undertaken on 109R and 134R for this reason. (Lines 129-130, Line 134).

Ln 209. Please reference Fig. 7, where the sea ice cover (a.k.a. concentration in this instance, for Walsh et al.) data are used.

We have referenced Fig. 7 as suggested.

Section 4.5 beginning Ln 293 on "Sea ice cover observational record":

Given the uncertainty in the Walsh et al. record, it might be good to run the change detect routines for Aug and Oct just to see if there is a material change in results, although perhaps this is unnecessary given that 5-yr running means are used.

We thank the reviewer for this comment. It is important to note that the models were run on the annual data, not the 5-year averages. We have presented the 5-year data sea-ice data for better comparison to the biomarker data, however we acknowledge that this is not clear in the methodology. We have updated this accordingly to make it clearer for the reader.

We have examined the Aug and Oct sea-ice concentration data and run the change point analysis, comparing the results to the September data currently used in the manuscript. The results are shown in the figure below, together with the various change points (vertical lines) associated with each station and month.

The long-term average observational sea-ice dataset from August, October and September shows that sea-ice concentrations in August and September are similar at all three sites. The October data is different from the trends in sea-ice concentration that we see in September and August for the sites, and this makes sense as we expect the sea-ice to begin to grow from this point.

[Figure]

Results of the change point analysis on the same observational sea-ice dataset is shown in the table below. There is a consistent change-point seen around 1970, but the shift at 1984 is less widespread across the different stations and months, as is evident from the September change-point analysis alone. It is only visible at 134R in September and October but not in August and is not distinguished when the averaged area or inner stations are examined. However, we believe that using the 1984 change is still valid based on the changes in mean and trend, and the consistency in the major changes no matter which of August, September or October were used.

| Core | Month of sea-ice data | Year of change | Type of change (model best fit – AIC) |
| --- | --- | --- | --- |
| DA17-NG-ST08-090R | August | 1958 | Mean cpt |
| | | 1964 | |
| | | 2016 | |
| | September | 1918 | Mean cpt |
| | | 1932 | |
| | | 1957 | |
| | | 1971 | |
| | | 2017 | |
| | October | 1919 | Mean cpt |
| | | 1931 | |
| | | 1936 | |
| | | 1942 | |
| | | 1960 | |
| | | 1971 | |
| | | 2017 | |
| DA17-NG-ST10-109R | August | 1953 | Mean cpt |
| | | 1959 | |
| | | 1966 | |
| | | 2017 | |
| | September | 1882 | Mean cpt |
| | | 1888 | |
| | | 1958 | |
| | | 1963 | |
| | | 1971 | |
| | | 2017 | |
| | October | 1919 | Mean cpt |
| | | 1931 | |
| | | 1936 | |
| | | 1942 | |
| | | 1963 | |
| | | 1971 | |
| | | 1978 | |
| | | 2000 | |
| | | 2017 | |

| DA17-NG-ST12-134R | August | 1952 | Trend AR (1) |
|---|---|---|---|
| | | 2016 | |
| | September | 1895 | Mean + AR (2) |
| | | 1984 | |
| | | 2015 | |
| | October | 1919 | Mean cpt |
| | | 1924 | |
| | | 1931 | |
| | | 1936 | |
| | | 1942 | |
| | | 1978 | |
| | | 1984 | |
| | | 1995 | |
| | | 2017 | |

As the September sea-ice data reflects the sea-ice minima it is arguably the best dataset to compare with biomarker data. Furthermore, the changes in August and September are very similar (see table above), again reflecting the sea-ice melt which is recorded in the biomarker data. The trends in the October sea-ice concentration do not match well with the biomarker data, likely because it marks the beginning of sea-ice accumulation. The change-point analyses for different months reveals little shift in the timing of changes in sea-ice concentration, especially when the five-year averages are considered, as the minor variations in the timing of key changes are within the five-year averages.

Thus, we continue to use the September sea-ice data for this study as we believe it best reflects the biomarker data. This justification is outlined in the methods (Lines 220-223).

Paragraph beginning Ln 390:

Please rewrite this sentence: "The positive correlations between IP25 and brassicasterol, and IP25 and dinosterol at all three sites (Fig. 5) can best be explained by the fact that under more extreme sea-ice conditions, both biomarkers show low values but with decreasing sea-ice, indicating more open-water and ice-edge conditions. "

Here is what I think is meant, but I am not sure about it:

"…can best be explained by the fact that when sea ice is preset more of the time, both biomarkers show low values. When open water conditions prevail, because the concentration of sea ice is low, or the ice edge moves shoreward of the location, both biomarkers show higher values. "

We have changed the wording accordingly to reflect the reviewers' comments and improve the clarity of this statement:

"The positive correlations between $IP_{25}$ and brassicasterol, and $IP_{25}$ and dinosterol at all three sites (Fig. 5) can best be explained by the fact that when more extensive sea ice is present both biomarkers show low values. When more open water conditions prevail, because the

concentration of sea ice is low, or the ice edge moves shoreward of the location, both biomarkers show higher values."

Ln 396. Suggest referring to Fig 4 here.

We have now referred to this in the text.

Ln 415. Consider replacing "seasonal sea ice has" with "areas of open water have" in this sentence: "The presence of IP25 in most of the samples in 90R and 109R suggests that seasonal sea ice has been present for the last ~120 years in the coastal and mid part of the Belgica Trough.

Seasonal sea ice generally refers to broad expanses where ice forms in the winter and melts or moves out in the summer, "an area of ocean that extends from the permanent ice zone to the boundary where winter sea ice extent is at a maximum; here, sea ice is present only part of the year; this zone primarily consists of first-year ice." (from the NSIDC glossary). Here, I believe you're referring to what biomarkers are indicating could be a fairly regular occurrence of polynyas in an area that is more often thought of as ice-covered.

Ln 419. Same comment as for ln 415, although in this sentence, could you replace "absence of seasonal ice" with "presence of sea ice" or "absence of periods of open water"?

We agree with the reviewer about the changes to this terminology in the above two sentences. This has been changed accordingly in the manuscript (Line 441 and Line 446).

The sea ice edge in this region, as defined using satellite passive microwave data, retreated north of the Belgica Bank area in 2021 (see https://nsidc.org/arcticseaicenews/2021/09/ ) but I believe that may have been only time that has happened in the satellite record.

Technical corrections

All of the technical corrections outlined below have been updated in the manuscript accordingly.

In Table 1, the longitude in the last row is missing a minus sign.

Ln 375 There is a missing "are".

---

## Referee Report (RR1)

Review of Manuscript: egusphere-2023-2363

Title: 120 years of sea-ice cover on the Northeast Greenland continental shelf: a biomarker and observational record comparison

by Joanna Davies et al. 2023

Thank you again for considering my comments and suggestions. Below some final comments before the manuscript may be acceptable for publication

1. In your revised manuscript you state that "you use organic bulk parameter data only to normalize the sea-ice biomarker data in this study". I think you should consequently do so. Remove Figure 3 and adjust chapter 4.2. Also remove lines 356-359 as you use XRF data only for "precise core correlation".
2. Check sentence in line 352 starting with "However, the reversal…" I think the sentence is incomplete.
3. Line 298: Should it be "The ratio of Bra/24-Me is high in the lower part of core 134R….?
4. Line 308: "Degradation products gradually increase from 8 cm in…

-

---

## Author Response (AR2)

**Response to Editor**

Thank you for your last revision of your manuscript "120 years of sea-ice conditions on the Northeast Greenland continental shelf: a biomarker and observational record comparison", submitted to EGUSphere/The Cryosphere, the improvements you and your co-authors made, and for additional information about your changes.

Thank you for the final comments on our manuscript, we have addressed all the changes in red below.

I kindly ask you to revise your manuscript according to the new comments made by reviewer 1 (see reviewer 1 report from 22 April 2024); in addition I have the three following comments:

(1) You write in the title and text in the beginning of the manuscript "120 years" and "recent changes (120 years)". In Fig. 8 the time axis/data ranges from 1880 to about 2015-17 (depending which subdiagram one looks at). I suggest (unless I have overseen it somewhere) to consider writing early in the manuscript from when to when your study applies (the term "recent" alone would make the statement somehow connected to a publication date), and adjust the total time amount, if it should be more than 120 years (like 130 or 135 years).

Thank you for pointing this out, we have now changed this to 130 years to reflect the 1880-2017 study period throughout the manuscript and in the title as suggested. As suggested, we have clarified the time period when we use the term 'recent' in the abstract and in more detail in the introduction (Lines 71-72). The reason for originally using 120 years is that that is our lowest reliable age, but of course, this is clear from the age model in the manuscript and 130 years is more correct.

I see that reviewer 2 had commented in the previous round (last comment before technical comments) the sea ice situation in 2021, and I saw no response to you on that point; possibly because this is after the time frame you discuss in your manuscript? I would appreciate if you could explain that.

Thank you for this comment, as you mention we do not discuss the sea ice changes after 2017 as it is outside of the study period constrained by the biomarker data due to the date of core collection. However, we agree that this is important to clarify so have added this to the methodology (Lines 218-220).

(2) In the conclusions in line 473 you write "until 1900" for what I understand means back in time. Two lines further, you use for another core the wording "prior to 1954". I find "prior to" clearer and easier to follow/understand, and I suggest to use another wording than "until" in line 473.

We agree that this is unclear, and the statement refers to data later than 1900 in cores 109R and 134R as this is where we have the first age marker. As there is no 1900 marker in 90R we use 'prior to'. We have updated the text to:

"For cores 109R and 134R, the age model is reliable for the time period after 1900, where the earliest age marker is found (210Pb reaches base level)."

Now lines 461-462.

(3) Recently, Wekerle et al. (2024; https://doi.org/10.1038/s41467-024-45650-z) published about Atlantic water warming of the ice tongue of the 79N glacier to the west of the area you are discussing. I suggest to consider taking this publication into account when discussing Atlantification/reasons for the regional sea ice changes (lines 460, 491 in version 3 of your manuscript), and where you introduce the 79N glacier and NEGIS (lines 90-99).

Thank you for highlighting this new paper, we agree that it should be added to our paper. This has been included in the regional setting outline (Lines 98-99) and in the discussion (Line 452) as suggested.

**Response to Reviewer 1**

Thank you again for considering my comments and suggestions. Below some final comments before the manuscript may be acceptable for publication

We thank the reviewer for taking the time to review our manuscript again and have responded to the specific comments below.

1. In your revised manuscript you state that "you use organic bulk parameter data only to normalize the sea-ice biomarker data in this study". I think you should consequently do so. Remove Figure 3 and adjust chapter 4.2.

We agree that removing this would make the purpose of this data in the manuscript clearer, so have done so accordingly. However, we believe that this dataset is still useful to provide in the supplementary material so have moved the figure to this part.

Also remove lines 356-359 as you use XRF data only for "precise core correlation".

This has been done.

2. Check sentence in line 352 starting with "However, the reversal…" I think the sentence is incomplete.

Thank you for noticing this mistake, we have now updated the text.

3. Line 298: Should it be "The ratio of Bra/24-Me is high in the lower part of core 134R….?

Yes, this was a mistake and we have updated with the reviewer's correction.

4. Line 308: "Degradation products gradually increase from 8 cm in…

We have corrected this in the manuscript.

---

## Author Response (AR3)

Dear Joanna Davies,

thank you for the last revision of your and your co-author's manuscript "130 years of sea-ice conditions on the Northeast Greenland continental shelf: a biomarker and observational record comparison", and for your responses to reviewer and editor comments. I appreciate very much all your work put into this, and I am happy to accept it for publication in "The Cryosphere", subject to a few, mainly technical corrections, see as listed in the following:

Thank you for accepting our manuscript, we are very grateful for the time spent reviewing our work. We have gone through all the notes below and responded in red.

List of comments (all line numbers refer to the ATC3 version of the manuscript):

• Lines 1, 16, 19, 73: The updated description of the time span the manuscript addresses; I appreciate that the time span description was adjusted, but I see still some inconsistencies with that. If you round to full 10er numbers it would be 140 (but I can understand that you did not chose that since it could be understood as an exaggeration). I suggest using in the title something more descriptive: "Sea ice conditions from 1880 to 2017 on the … ". In the abstract you could write "137 years" instead of "130 years", and "137-year study period". At the end of section 1, I suggest writing "to the last ca. (or tilde symbol) 140 years, spanning from …", because here you connect to now (2024), which is 144 years after 1880.

Thank you for raising this point, we have updated the title of the manuscript according to your suggestions. However as the age constraint on sediment cores, and thus the biomarker data is somewhat uncertain we have removed reference to 130/137/140 years in the abstract.  We agree that putting the number to 140 is an exaggeration of the study period length in this instance.

As the dataset extends only to the date of sediment core collection, 2017 rather than 2024, we have kept the numbers the same at the end of section 1.

• Line 4: A comma is missing after "Ruediger Stein" in the author list.

Corrected

• Line 8: The country for the address of affiliation #4 is missing.

Corrected

• Line 126: The paragraph on samling (3.2) does not contain information on any samling on core 92G, but later (lines 137 and 216), grain size and Hg content analyses on this core are mentioned, respectively. I suggest adding some brief information in 3.2.

• Line 139: I suggest adding the country after Aarhus University.

Corrected

• Line 139: I suggest writing (NaPO3)6 with lower 3 and 6 as in other cases for chemical formulas.

Corrected

• Line 216: You wrote earlier (line 120) you would just use 92G when addressing this core in the further text. So, you could avoid the full name of this core here.

Corrected

• Line 374: I suggest adding a comma after "dramatic".

Corrected

• Line 442: I suggest to here write "landfast" in one word.

Corrected throughout the manuscript

• Line 494: Instead of the tilde symbol I suggest writing "about" ahead of "1970". I assume "about" represents what the authors want to express. Tilde is also used several other places in the manuscript. Usually it fits, but in the conclusion statement I would find using "about" more suitable.

Corrected

• Line 509: I suggest adding the country after Plymouth.

Corrected

• Line 524: There are some technical issues with the reference Ananicheva et al. 2011: For the SWIPA 2011 report (https://www.amap.no/documents/doc/snow-water-ice-and-permafrost-in-the-arctic-swipa-climate-change-and-the-cryosphere/743), I have seen other citation forms, such as "AMAP, 2011", and for any subchapter of the report one may choose the authors and title of that. After the author list (which I cannot connect with the author listing to the entire SWIPA 2011 report or a specific part of it), three words in Russian meaning "Publication title, page, citation" are listed, and after that, not a full publication title is listed - some text seems to be missing here.

Corrected

 Note also that in the author list, a comma is missing after "Van Oort", and the connected initials "B. E. H." should all be capitalized. Please check what was supposed to be cited, and correct accordingly.

Corrected

Thank you again for your efforts with this manuscript.

Regards

Sebastian Gerland